# Autophagy and Extracellular Vesicles, Connected to rabGTPase Family, Support Aggressiveness in Cancer Stem Cells

**DOI:** 10.3390/cells10061330

**Published:** 2021-05-27

**Authors:** Aude Brunel, Gaëlle Bégaud, Clément Auger, Stéphanie Durand, Serge Battu, Barbara Bessette, Mireille Verdier

**Affiliations:** EA 3842-CAPTuR, GEIST Institute, Faculty of Medicine, University of Limoges, 2 rue du Dr Marcland, CEDEX, 87025 Limoges, France; aude.brunel@unilim.fr (A.B.); gaelle.begaud@unilim.fr (G.B.); clement.auger@unilim.fr (C.A.); stephanie.durand@unilim.fr (S.D.); serge.battu@unilim.fr (S.B.); barbara.bessette@unilim.fr (B.B.)

**Keywords:** rab family, autophagy, extracellular vesicles, cancer stem cells

## Abstract

Even though cancers have been widely studied and real advances in therapeutic care have been made in the last few decades, relapses are still frequently observed, often due to therapeutic resistance. Cancer Stem Cells (CSCs) are, in part, responsible for this resistance. They are able to survive harsh conditions such as hypoxia or nutrient deprivation. Autophagy and Extracellular Vesicles (EVs) secretion are cellular processes that help CSC survival. Autophagy is a recycling process and EVs secretion is essential for cell-to-cell communication. Their roles in stemness maintenance have been well described. A common pathway involved in these processes is vesicular trafficking, and subsequently, regulation by Rab GTPases. In this review, we analyze the role played by Rab GTPases in stemness status, either directly or through their regulation of autophagy and EVs secretion.

## 1. Introduction

Oncology has been a widespread field of research in the last few decades. Despite many advances, there are still some problems to solve. Indeed, in some cases, relapses are frequently observed, linked to resistance to therapeutic treatments. This is, in part, due to a particular cancer cell population called Cancer Stem Cells (CSCs). These represent a small part of the tumor mass: 0.5–5% growing up to 10% in some cases. First discovered in leukemia [1,2], their existence has been rapidly demonstrated in solid tumors such as breast cancer [3] and brain cancer [4] and nowadays, in almost all cancers. CSCs share with physiological stem cells self-renewal, unlimited proliferation rate and multipotent capacities. The first characterization of CSC relied on their ability to regrow a tumor *in vivo* [1]. However, even today, CSCs remain difficult to characterize and isolate. Based on the same rationales as for other cellular subpopulations, CSCs are thought to be identified by the expression of specific markers. Indeed, membranal proteins such as CD133, CD44, LGR5 or intracellular actors—mainly transcriptional factors, i.e., BMI1, Oct4, Nanog, sox family—were described to be enhanced in stemness [5,6]. Nevertheless, unlike for other cellular subsets, it is not possible to establish a solid link between CSCs and unequivocal stemness markers. This explains why scientists also consider functional properties to define this peculiar population. Therefore, clonogenic faculty, chemotherapeutic resistance, metastatic propension [7] as well as quiescent stage and drug efflux are also commonly evaluated to identify CSCs [8]. However, these properties do not really define if these cancer cells are really cancer stem cells, progenitors or cancer stem-like aggressive cells. It is important to be aware that in many studies, including those cited in this review, the tumorigenicity requirement, which should be unavoidable, is not always verified. Readers, by referring to the cited publications, will be able to make up their own minds of stemness. Recently, the concept regarding CSCs has been evolving and no longer considers CSCs as a static entity, but rather as a continuum, constantly sprouting and adapting to changes in the microenvironment [9]. Thus, instead of having a unique CSC clone, tumors are composed of several CSC microstates, reflecting the high heterogeneity of the tumor. It appears that there is a dynamic reversibility between non-stem cell and stem cell states, which makes CSCs even more complicated to understand and target. CSCs are indeed highly regulated, including through the microenvironment. For example, it has been shown in breast and prostate cancer cell lines that IL-6 secretion may tip the balance in favor of a “stem-like cell” phenotype [10].

It is now well known that various processes are involved in the maintenance and status of CSCs. Among them, we decided to focus here on two particular processes, playing a key role in physiological as much as in pathological mechanisms: autophagy and Extracellular Vesicles (EVs) secretion. Three major types of autophagy have been described: micro-autophagy, chaperone-mediated autophagy and macro-autophagy. The last one, which we will be focusing on in this review, is commonly known as “autophagy”. It is a highly conserved degradation and recycling mechanism of cellular components, complementary to the proteasome. The autophagic process has an important role in the maintenance of cellular homeostasis and any dysfunction can easily lead to several pathologies, including cancer [11].

The second cellular process this review is focusing on is the secretion of Extracellular Vesicles (EVs). Among them are apoptotic bodies, microvesicles and exosomes. The last ones are nanovesicles secreted by a wide variety of cellular types, including tumor cells. They support tumor aggressiveness through the transfer of their content, thus changing the phenotype and/or behavior of the recipient cells. As an example, we showed in a previous study [12] that transfer of surface receptor TrkB (Tropomyosin receptor kinase B) by the secreted EVs of glioblastoma cells led to a restored aggressive phenotype of the non-aggressive shChi3L1 cell line. Furthermore, as EVs can easily be detected in many body fluids [13,14,15,16], some studies are pointing their advantages out as diagnosis and prognosis markers by performing a simple liquid biopsy. Indeed, in many cancer types, a difference in EV content between healthy persons and patients with cancer has been observed [17,18,19,20].

Autophagy and EVs secretion have clearly common points such as the involvement of the lysosome or their activation under stress conditions. Moreover, those two processes include vesicular trafficking, which means that Rab small G protein family is required for each of them. Rab GTPases are small G proteins belonging to the Ras superfamily. As with their counterparts, they balance between an active state, GTP-binding, and an inactive state, GDP-binding following GTP hydrolysis. To ensure this balance, they need the intervention of two factors. Indeed, the switch between GDP and GTP is performed by Guanine-nucleotide Exchanged Factors (GEFs), while GTP hydrolysis is amplified by GTPases’ Activated Proteins (GAPs). Over 70 Rab GTPases are described, each of them being able to interact with different effectors. Therefore, they are considered as markers of cellular compartments, since at least one of them is specific to each compartment (Figure 1). There is a considerable amount of evidence showing the involvement of Rab GTPases in cancer progression [21,22,23,24] but very little concerning their role in cancer stemness.

In this review, we focused on the role played by autophagy and EVs secretion in CSCs status and how Rab GTPases are acting in the triumvirate. Indeed, there are very few recent studies showing a direct role of this family in stemness, which will be explained.

## 2. Rab Family, Autophagy and CSCs

Autophagy is a mechanism that consists of the cytoplasmic formation of a double membrane vesicle, containing long-lived or damaged components such as proteins, lipids, organelles. This vesicle, called the autophagosome, will ultimately fuse with a lysosome, hence leading to the degradation of its content into elementary entities (amino acids, fatty acids, sugars, etc.). This is a way of providing nutrients and energy to the cell [27,28]. The formation of the autophagosome is a multistep process, in which several protein complexes are necessary for phagophore initiation (ULK1/2 complex), nucleation (Beclin-1 complex) and elongation (ATG5/12/16 and MAP-LC3 proteins) (for a review, see [11]). Whereas a family of almost 30 ATG (autophagy-related gene) proteins is involved, the main one is Microtubule-associated protein 1A/1B-light chain 3 (MAP-LC3) which allows closure of the vesicle. This protein is also used as a marker of the autophagic flux (fusion between autophagosome and lysosome).

As autophagy involves vesicular trafficking, it is not surprising to find Rab GTPases as essential actors. Among the Rab GTPases family, there are mainly 10 Rabs involved in the autophagic process [26]. Most of them act at the autophagosome level. One of the first Rab-GTPase described in autophagy regulation was Rab33B in 2008, a Golgi-associated Rab protein [29] via its interaction with the ATG16L protein. This protein is one of the components of a protein complex involved in autophagosome elongation. Additionally, Rab7 might be one of the most described Rab-GTPase protein in autophagic pathway regulation. Indeed, this particular protein plays an important role in lysosome biosynthesis and function [30]. This is why autophagosome fusion with the lysosome has been, in part, attributed to Rab7, particularly to Rab7a [31]. Moreover, Rab1 seems to participate in phagophore formation by regulating Atg9 localization [32,33], whereas Rab32 is required for autophagosome formation [34,35] and Rab5 for its closure [36]. In addition, Rab24 has been shown to be co-localized with MAP-LC3 [37]. Furthermore, Rab2 has recently been shown to be involved in the regulation of autophagosome and autolysosome formation in different mammalian cell lines [38]. A 2019 study on colon cancer cell lines demonstrated that the inhibition of Rab5/7, both playing a role in the endolysosomal pathway, could decrease the CSCs compartment [39]. Authors used mefloquine (MQ), an antimalarian drug, as an inhibitor of Rab5/7. They highlighted an impairment of mitophagy, the specific recycling of mitochondria via the autophagic pathway, and lysosomal activity inhibition by suppressing LAMP1 and LAMP2 expressions (lysosomal-associated membrane protein) both *in vitro* and *in vivo* using mouse PDX models. The combination of MQ with conventional colon cancer drugs led to a drastic decrease (down to 9.4%) in CD44v9^+^/CD133^+^ CSC population after MQ treatment. This effect was even stronger with the combined treatments and has also been shown in PDX models by an IHC staining of CD44v9^+^.

Autophagy has a critical role in embryonic development and is a key actor of cellular homeostasis, which is why its deregulation can easily lead to various diseases, including cancer [40]. Additionally, growing evidence is demonstrating the essential role of autophagy in physiological stemness in many cell types [41,42,43,44,45] as well as for CSCs maintenance and function. Nevertheless, the role played by autophagy in cancer is still not completely understood, as there is evidence of a pro- and anti-tumor autophagy, presuming a duality of the autophagic role in cancer [46,47]. Indeed, autophagy has been described as a cytoprotective mechanism against many cellular stresses that could be considered as an onco-suppressor, but it also protects cancer cells from their hypoxic environment, nutrient deprivation and treatments [48,49]. This is why we decided to investigate, on the one hand, autophagy as a “pro-CSCs” survival mechanism and, on the other hand, autophagy as an “anti-CSCs” mechanism, as there are few studies supporting the idea that autophagy can lead to CSCs differentiation.

The autophagic flux in the maintenance of CSCs stemness and function has been widely explored, in particular in Breast Cancer Stem Cells (BCSCs) [50,51,52,53]. This has particularly been demonstrated through studies on various autophagy proteins such as Beclin1 [54], Atg4A [55], implicated in MAP-LC3 maturation, or using drugs such as chloroquine or salinomycin to inhibit the fusion between the autophagosome and lysosome, i.e., autophagic flux [56,57]. Evidence of the implication of autophagy in CSCs survival for many other cancer types has also been given [58,59,60,61]. This is notably true for bladder cancer in which autophagy has been demonstrated as essential for stemness maintenance of bladder CSCs and for tumorigenic properties such as invasion and metastasis [62,63]. Autophagy-promoted metastasis is also known in gastric cancer [64]. Indeed, the Atg4A protein is shown to be responsible for inducing the epithelial–mesenchymal transition as well as stemness properties in various gastric cancer cell lines and *in vivo* using mice models. As previously mentioned, CSCs are key actors in therapeutic resistance, which is why numerous studies on different types of cancer have proven the beneficial effects of autophagy inhibition on therapy sensitization [50,65,66,67,68,69,70,71,72]. It has notably been demonstrated in non-small cell lung carcinoma (NSCLC) [67]. An enhanced CD133^+^ cell population after chemotherapeutic treatment using cisplatin has been shown both *in vitro* in the A549 cell line and ex vivo in clinical NSCLC samples. A higher autophagy level in those CD133^+^ cancer stem cells has been highlighted, which led the authors to try to inhibit this particular process in order to target the stem cells population. By using chloroquine, a loss in CD133^+^ cells and a decrease in sphere forming abilities in A549 cells were observed. Furthermore, a combined treatment using cisplatin and chloroquine showed a synergistic effect, inducing a decrease in tumor growth in engrafted NOD/SCID mice. Similar observations had already been made a few years earlier in ovarian cancer stem cells (OCSCs) [70]. A high level of autophagy has been shown is OCSCs. Furthermore, blocking the autophagic flux either by using chloroquine or knocking down ATG5 expression reduced the stemness capacities both *in vitro* and *in vivo*. Additionally, autophagy impairment increased chemotherapy sensitivity by significantly impacting cell viability *in vitro* as well as *in vivo* and by drastically reducing tumor weight and volume.

Aside from its pro-tumoral effects through CSCs maintenance and function, since a few years ago, autophagy has also been attributed a role in CSCs differentiation and thus, their sensitization to chemotherapeutic treatment. Some even point out the beneficial role of curcumin in inducing autophagy-mediated differentiation [73,74]. In the peripheral nervous system, Li and colleagues [75], working on neuroblastoma cell lines, showed that calcium/calmodulin-dependent protein kinase II was responsible for autophagy activation through Beclin1 phosphorylation. Furthermore, there was an induction of Id—inhibitor differentiation—degradation through autophagy, leading to cell differentiation. This has also been highlighted in the central nervous system and particularly in glioma-initiating cells [74,75,76,77,78,79,80]. Furthermore, Ciechomska and co-workers [78] observed autophagy activation using an inhibitor of Histone Methyl-Transferase (HMT), BIX01294, on Glioblastoma Stem Cells (GSCs). Moreover, after this treatment, an increase in astrocytic (GFAP; Glial Fibrillary Acidic Protein) and neuronal (TUBB3; Tubulin Beta 3 class III) differentiation markers in GSCs was revealed. This autophagy-mediated differentiation has been highlighted in some other cancer types, including colon, liver and gastric cancers [81,82,83]. A 2019 study [84] demonstrated both *in vitro* and *in vivo* that metformin, a common drug used to treat type II diabetes, could suppress the self-renewal and tumorigenicity abilities of Osteosarcoma Stem Cells (OSCs). Furthermore, metformin was shown to induce a cell cycle arrest in OSCs cell lines, as well as Reactive Oxygen Species (ROS)-mediated apoptosis and autophagy.Treated OSCs were impaired in their capacity to form spheres and showed a significant decrease in stemness markers Oct4 and Sox2. Finally, treated OSCs transplanted in mice led to a significant decrease in both tumor weight and volume. Then, a 2020 study also found out a link between ROS-mediated autophagy and proliferation and stemness status by working on colorectal cancer cells [85]. They investigated the effects of silencing LETM1 (Leucine zipper-EF-hand-containing Transmembrane protein 1), which is overexpressed in CRC tissues compared to normal ones and of poor prognosis. They first found a decrease in colony forming and proliferation capacities, in addition to an accumulation of S-phase cells. The monitoring of autophagy in esi-LETM1 cells showed a significant increase in Beclin1 expression and MAP-LC3II/I ratio, meaning an enhanced autophagic process. Few years earlier, Sharif et al. [86] worked on PHosphoGlycerate DeHydrogenase (PHGDH), previously shown as required for CSCs maintenance in hypoxia-induced Breast Cancer Stem Cells (BCSCs) [87]. They observed a significant positive correlation between PHGDH and Oct4 expressions both *in vitro* in Embryonic Carcinoma Stem-Like Cells and in Cancer Stem-Like Cells from patients. Therefore, they decided to undertake a *PHGDH* knockdown (KD) and demonstrated that it decreased the stemness characteristics and promoted Embryonic Carcinoma Stem-Like Cells differentiation in multilineage. Moreover, by looking for different autophagy proteins’ expression, they revealed an activation of autophagic flux in cells KD for *PHGDH*.

To summarize, autophagy has been widely studied for a few decades, but its involvement in cancer and CSCs remains uncertain. Indeed, in some cases, it appears that inhibiting autophagy impairs CSCs compartment and increases therapy sensitization, whereas in some others, inducing autophagy seems to be a better strategy to promote CSCs differentiation. Taken together, those data suggest that, although cancer therapy research is likely focused on finding anti-cancerous molecules that could be used in any type of cancer, concerning autophagy-targeted treatments, it might be of better interest to consider a cancer-dependent treatment.

## 3. Rab Family, Extracellular Vesicles and CSCs

Extracellular vesicles (EVs) include apoptotic bodies, microvesicles and exosomes. The last ones’ secretion is initiated by invagination of the endosomal membrane, leading to Multivesicular Bodies (MVBs) formation containing Intraluminal Vesicles (ILVs). They are finally targeted to the plasma membrane, fuse with it and release the ILVs into the extracellular environment. These exosomes are secreted with a size comprising between 30 and 150 nm (for review see [88]). EVs reflect the physiological status of the secretory cell [89] through their varied content, which includes proteins such as surface receptors, nucleic acids, lipids, etc. They are of major importance for cell-to-cell and cell-to-microenvironment communications.

As the secretion of Extracellular Vesicles (EVs) involves the endosomal pathway, it is not surprising that Rab GTPases are essential to this process. There are mainly three Rab-GTPases described in EVs biogenesis and secretion [90]: Rab11, Rab35 and Rab27A. All three proteins are implicated in the docking and fusion of the MVBs. Rab11 activity has been shown to be calcium-dependent in the erythroleukemia cell line K562 in 2005 [91]. More recently, the promotion of exosome secretion by Rab35 has been observed in hepatocellular carcinoma [92], when it had been previously demonstrated few years earlier in Oli-neu cell line, i.e., oligodendroglial precursor [93]. Although, the most studied and commonly known is Rab27A/B, also involved in the docking and fusion of MVBs at the plasma membrane [94]. One of the first studies showing the role of Rab27A in EVs secretion was carried out on HeLa cells [95] and then, confirmed in breast cancer cells [96]. There are some studies on the expression of Rab27A in different cancer types and its prognostic relevance. A high expression appears to be of poor prognosis in pancreatic cancer [97], bladder cancer [98], hepatocellular carcinoma [99] and gliomas [100]. This is also the case concerning an overexpression of Rab27B in non-small cell lung carcinoma [101]. On the contrary, for colon and prostate cancer, it seems that a high expression of Rab27A or Rab27B is correlated with a better outcome [102,103].

EVs secretion is an essential mechanism for cell-to-cell communication and cell-to-microenvironment communication. This is notably true concerning CSCs, which are permanently communicating, modulating and adapting to their microenvironment. Even though stem cell-derived EVs are studied as therapeutic tools in cancer [104,105] and also in other kinds of diseases [106,107,108,109] or even as regenerative tools [110], little is known about the action mechanism of EVs on CSCs maintenance. EVs have been attributed an emerging role of transfer of nucleic acids, such as miRNAs [111,112,113,114] or lncRNAs [115,116] and particularly for their involvement in stemness status. Ren et al. [115] worked on colorectal cancer (CRC) and the potential transfer of long non-coding RNA (lncRNA) H19, known to be of poor prognosis in CRC [117]. They found that H19 is overexpressed in CRC patients and particularly in Carcinoma-Associated Fibroblasts (CAFs). Using siRNA targeting H19 in CRC cell lines SW480 and HCT116, a significant decrease in the number of spheres and sensitization to oxaliplatin treatment was observed. A higher level of H19 was exhibited in patients’ CAF exosomes compared to normal fibroblast exosomes, thus imputing the stemness capacities to a transfer of H19 via exosomes secreted by CAFs. In 2017, Figueroa and colleagues [113] investigated the role played by exosomes from Glioma-Associated human Mesenchymal Stem Cells (GA-hMSC) on Glioma Stem Cells (GSCs). They observed *in vitro* an increase in clonogenic capacities of GSCs and *in vivo* an increase in the GSCs’ tumorigenicity. Among GA-hMSC exosomes’ miRNAs, miR-1587 was found to be responsible for the proliferation and self-renewal abilities of GSCs by targeting the tumor suppressor NCOR1 (Nuclear hormone receptor Co-repressor 1). Furthermore, another team worked on GSCs secreted exosomes and their mode of action on non-GSCs in several glioblastoma cell lines [118]. Indeed, those two cellular populations coexist in the tumor mass and understanding the way they communicate is as challenging as it is interesting to find new treatments. A neurosphere formation assay revealed colonies of higher number and size after GSC-derived exosomes treatment. Enhanced expression of several stemness markers in non-GSCs treated with GSC-derived exosomes was also highlighted, meaning a stemness transfer in the recipient cells. This has been confirmed in a mouse model by injecting glioblastoma cells treated or not with GSC-derived exosomes. When treated, the cells promoted the formation of bigger and heavier tumors. Afterwards, Notch1 transfer via GSCs derived exosomes into non-GSCs cells has been demonstrated to be responsible for transferring stemness capacities and inducing dedifferentiation. Then, a 2020 study on group 4 medulloblastoma primary cultures showed that miR-135a- and -135b-containing EVs could regulate stemness [119]. Tumor tissues were enzymatically dissociated to form single cells and to be cultured into Bulk Tumor Cells (BTCs) and Brain Tumor Spheroid-forming Cells (BTSCs) supposed to be enriched in stem cells and progenitors. Authors then performed a miRNA profiling on the cells and secreted EVs and the expression of numerous miRNAs was found to be significantly increased in BTSCs and BTSCs-EVs compared to the BTCs. Among them were miR-135a and -135b, which led to an impairment of the stemness capacities in BTSCs when downregulated. There is growing evidence on the impact of EVs on CSCs stemness and aggressiveness in several cancer types such as esophageal carcinoma [120] or pancreatic cancer [121,122]. Yan et al. [123] demonstrated that EVs derived from Lewis Lung Carcinoma could reprogram mouse-induced pluripotent stem cells into CSCs *in vitro*, endowing them with sphere formation abilities. Moreover, those converted CSCs were given tumorigenic and metastatic capacities in a nude mice model.

Some investigations described EVs as being responsible for inducing therapeutic resistance in CSCs by enhancing and maintaining stemness in cancer cells, mainly in colorectal cancer [111,124,125,126]. One of the most recent studies also investigated the CAF-secreted exosomes and their way of action on stemness and therapeutic resistance [111]. The transfer of miR-92a-3p from CAFs to cancer cells via exosomes allowed the promotion of resistance to the chemotherapeutic agent 5-FU (5-Fluorouracile), metastasis and epithelial to mesenchymal transition. An enhancement of colorectal cancer cells stemness was also demonstrated by performing plate colony formation and sphere formation assays. A previous study [126] had already shown the contribution of CAF-derived exosomes to the chemotherapy resistance of colon CSCs to 5-FU and oxaliplatin, thus leading to the idea of targeting CAFs before therapy to unravel the exosomes’ secretion. Another team recently demonstrated the potential of CAF-derived exosomes to participate in radiotherapy resistance by promoting the stemness abilities of colon cancer cells [127]. They succeeded in demonstrating a higher number of spheres and resistance to radiotherapy when cells are treated with those particular EVs. Furthermore, *in vivo* experiments showed a greater capacity of rapidly forming larger tumors after radiation when CRC cells are previously treated with CAF-derived exosomes. Furthermore, EVs secreted following environmental stresses such as chemotherapy [112] or hypoxia [128] are able to promote stemness in surrounding cancer cells. Indeed, Ramteke and colleagues worked on prostate cancer cell lines LNCaP and PC3 and investigated the role of exosomes secreted under normoxic (21% O_2_) or hypoxic (1% O_2_) conditions. It appeared that treating the cells with exosomes secreted under hypoxia led to a higher invasiveness, migration rate and an especially higher number of prostaspheres compared to the normoxic exosomes treatment condition.

Altogether, these data suggest that EVs are essential actors of CSCs maintenance. This is why targeting them could be a new therapeutic approach to prevent the transfer of stemness abilities and henceforth, resistance to treatment. A recent study [129] used engineered biological nanoparticles similar to EVs to treat hepatocellular cancer by targeting liver CSCs. It was also demonstrated a few years earlier in early-stage breast cancer cells that targeting EVs could be efficient [130]. One team also investigated a possible way to use EVs as therapeutic tools to reprogram CSCs and trigger their differentiation [104]. For this purpose, they used exosomes from osteogenic differentiation of human adipose-derived stem cells on CD133^+^ cells. They were able to observe the induction of an osteogenic differentiation in the recipient cells.

## 4. Rab GTPases and CSCs

Rab GTPases have been described in normal stemness in some studies [131,132]. For instance, Rab8a has been highlighted in intestinal stem cells [133,134], or Rab31 in neural progenitor cells [135]. There is not a lot of evidence concerning a direct link between Rab GTPases and CSCs status. However, the overexpression of Rab5b in breast cancer stem cells was highlighted indirectly in a 2018 study [136]. Indeed, authors showed that miR-130a-3p, involved in Rab5b regulation, is downregulated in such CSCs. Furthermore, some Rabs are at the crossroad with the role played by several Rab GTPases in EVs release, especially concerning Rab27A/B. A recent study realized by Peng and coworkers [137] showed that the specific targeting of Rab27B in order to interfere with exosomes secretion could eliminate acute myeloid leukemia (AML) stem cells. The miR-34c-5p has been shown to be downregulated in primary AML CD34^+^/CD38^−^ cells. This is why authors investigated the effect of miR-34c-5p overexpression. They were able to highlight an elimination of AML stem cells via inducing senescence and especially by decreasing Rab27B expression and thus, exosome release. Cheng et al. [24] also investigated the role of Rab27B in the stemness of two colorectal cancer cell lines. Colorectal cancer stem cells from HT29 and HCT15 cells were isolated and grown in a serum-free spheroid cultivation system to be enriched in stem-like and progenitor cells. Those cultures were called sphere-derived CSCs (SDCSCs). Silencing *Rab27B* led to a decrease in sphere formation of SDCSCs and to an attenuation of the tumor growth and CD44^+^ population in CD44^High^ CRC PDX. Then, miR-146a-5p was found to be increased in SDCSCs exosomes, which are allowed to be released via Rab27B. Authors demonstrated that downregulating miR-146a-5p was responsible for an impairment in the sphere formation and tumorigenic capacities first given by SDCSC-derived exosomes. Finally, a positive correlation between exosomal miR-146a-5p expression in patients’ sera and a stemness cellular profile was shown.

Concerning the importance of Rab27A in the stemness status of cancer cells, there was a 2016 study that focused on the role played by miR-134-3p in human ovarian cancer stem cells [138]. It appeared that this miRNA is able to bind to the 3′UTR (Untranslated Region) site of Rab27A mRNA, thus interfering with its expression. Ovarian Cancer Stem Cells (OCSCs) were treated either with a wild-type miR-134-3p or with a mutant, and CSCs markers’ expression was analyzed. Authors observed a significant decrease in the expression of those markers, with the wild-type condition corresponding to a loss of Rab27A. In a xenograft model, they were able to show an impairment of tumor growth when miR-134-3p is overexpressed. In 2018, Chano et al. [139] demonstrated the role played by Rab39A in cancer stemness. This isoform is localized at late endosomes and lysosomes [140] and regulates phagosomes acidification [141], whereas Rab39b is rather localized at the Golgi apparatus. The authors used shRNA targeting Rab39A in human osteosarcoma CSCs and showed an impairment of stemness capacities *in vitro* by performing a clonogenic assay. They observed a significant decrease in the size and number of colonies. *In vivo*, shRab39A impaired tumorigenesis in xenografted mice. Then, using RNA-seq analysis, they tried to identify a Rab39A downstream effector. It appeared that RXRB (Retinoid X Receptor Beta) could be a potential candidate. Indeed, in shRab39A cell lines, RXRB expression decreased and when authors induced an overexpression of *RXRB* gene, the ability of shRab39A CSCs to form spheres was restored. Finally, targeting *RXRB* expression by shRNA had the same effect on CSCs as shRab39A, showing the implication of the Rab39A–RXRB axis in the maintenance of CSCs stemness.

Another study showed that Rab6 could have a role in lung CSCs (LCSCs) sensitivity to cisplatin-based chemotherapy [142]. Rab6 is located at the Golgi apparatus and regulates the vesicular trafficking from the Golgi toward the endoplasmic reticulum (ER) [143]. Authors demonstrated that miR-5100 is responsible for LCSCs cisplatin resistance through the downregulation of *Rab6* expression. It seems that miR-5100 is upregulated in LCSCs compared to non-CSCs and, when overexpressed, increases CSCs properties. By looking at miR-5100 target genes, they found that Rab6 was downregulated in LCSCs. Furthermore, when Rab6 expression is increased using pcDNA plasmids, the expression of the common stemness markers was decreased. Another team worked on Breast Cancer Stem Cells (BCSCs) and focused on the prolyl isomerase Pin1, as it has been shown to be increased in human BCSCs, playing a key role in their promotion and maintenance [144,145]. They identified Rab2A as a downstream effector of Pin1. Authors highlighted that Rab2A is overexpressed in human cancers and thus, increases the BCSCs-enriched population. Finally, the study showed that the role of Rab2A in BCSCs is mediated by the activation of the Erk1/2 pathway and the translocation of β-catenin to the nucleus [146]. Furthermore, as previously mentioned, there are some publications showing growing evidence of the role of Rab27A, but also of its counterpart Rab27B, in CSCs status. Indeed, studies in BCSCs and colon CSCs [147,148] attributed to Rab27A a role in stemness maintenance. A recent study focused on Rab37 small G protein in lung cancer. Contrary to the other Rab GTPases quoted above, this one is known for its tumor suppressor action [149,150]. Authors investigated its potential role in LCSCs and observed that Rab37 nullifies LCSCs stemness via inhibition of the Wnt signaling pathway [151]. Indeed, downregulation of Rab37 expression seemed to enhance the stem-like properties of lung cancer cells both *in vitro* and *in vivo*. To do so, on the one hand, they used shRNAs targeting Rab37 and performed three-dimensional sphere culture assays and observed an increase in the size and number of the spheres. Furthermore, RT-qPCR results showed a significant increase in the expression of several stemness markers, which was confirmed in a mouse xenograft model. On the other hand, they overexpressed Rab37 by either inducing a constitutively active form of the protein or using a dominant negative mutant. The sphere formation assay and RT-qPCR results were in favor of a significant loss of stemness properties. In fact, Rab37 is responsible for the exocytosis of SFRP1 (Secreted Frizzled-Related Protein 1), which inhibits the Wnt pathway in LCSCs. By performing analysis in a lung cancer patients’ cohort, they showed that a low expression of Rab37, and thus of SFRP1, is associated with poor prognosis and high Oct4 expression, a well-known stemness marker.

In addition, CSCs exhibiting high capacities to undertake Epithelial-to-Mesenchymal Transition (EMT) support metastasis, and subsequently, recurrence and poor prognosis in many cancers. This process consists of phenotypic modifications, notably the loss of E-cadherin toward N-cadherin expression. Various Rabs have been attributed a role in cancer cells migration and Epithelial-to-Mesenchymal Transition (EMT). Among them, Rab23 was found to be responsible for EMT in ovarian cancer and hepatocellular carcinoma [152,153]. Additionally, Rab11 has been described as a regulator of E-cadherin, thus promoting cell migration in colorectal cancer [154,155]. On the contrary, some Rab-GTPases negatively regulate EMT and their silencing has been proven to induce it. This is notably the case concerning the epigenetic silencing of Rab39A in cervical cancer [156] or the downregulation of Rab17 in non-small cell lung carcinoma [157].

All of these studies, in addition to what is known concerning the role played by those small G proteins in autophagy, extracellular vesicles secretion and/or cancer stem cells status (Figure 2), demonstrate the Rab GTPases’ potential to be new therapeutic targets to modulate cancer stemness and make the current treatments, especially chemotherapies, more efficient.

## 5. Conclusions

This quick update on the current knowledge concerning autophagy, EVs and Rab GTPases in CSCs maintenance identifies new possible therapeutic strategies, especially via targeting Rab GTPases. Indeed, even if the question of their real tumorigenicity is still debated, their intervention in key stages of the life of a cell justifies the development of therapies targeting them. Furthermore, this has been the topic of a quite recent review [158]. This goal could be achieved by targeting of Rabs’ regulatory proteins [159], particularly their GEFs. A team designed nine peptides that could target Rab GTPases through their different interactions [160], which was performed by Mitra and coworkers in 2017 using stapled peptides inhibitors of Rab25 [161] in various cell lines. This might also be achieved by the means of chemical agents such as Nexinhibs (Neutrophil-exocytosis inhibitors), which have been found to be inhibitors of the interaction between Rab27a and its effector JFC1, i.e., synaptotagmin-like protein1 [162]. Thus, even if these innovative therapies targeting Rab proteins are promising, further studies would have to be undertaken. Indeed, depending on cell type, tumoral stage/grade and even the cancer type, the targeted Rab might not be the same, leading to personalized patient care.

## Figures and Tables

**Figure 1 cells-10-01330-f001:**
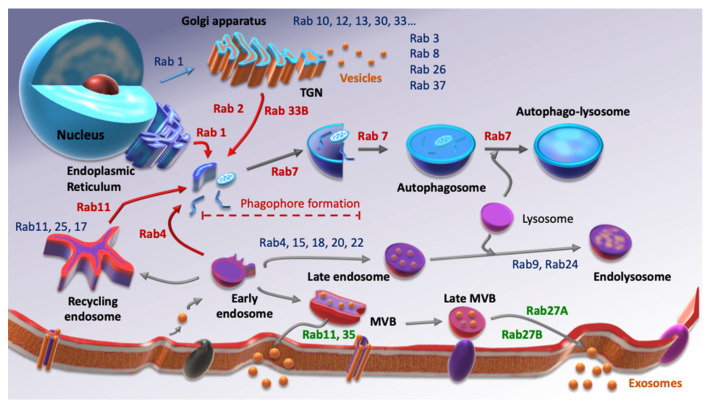
Screening of some Rab GTPases involved in autophagy (red), EVs secretion (green) and other Rab proteins involved in other cellular processes (blue). Inspired from [25,26].

**Figure 2 cells-10-01330-f002:**
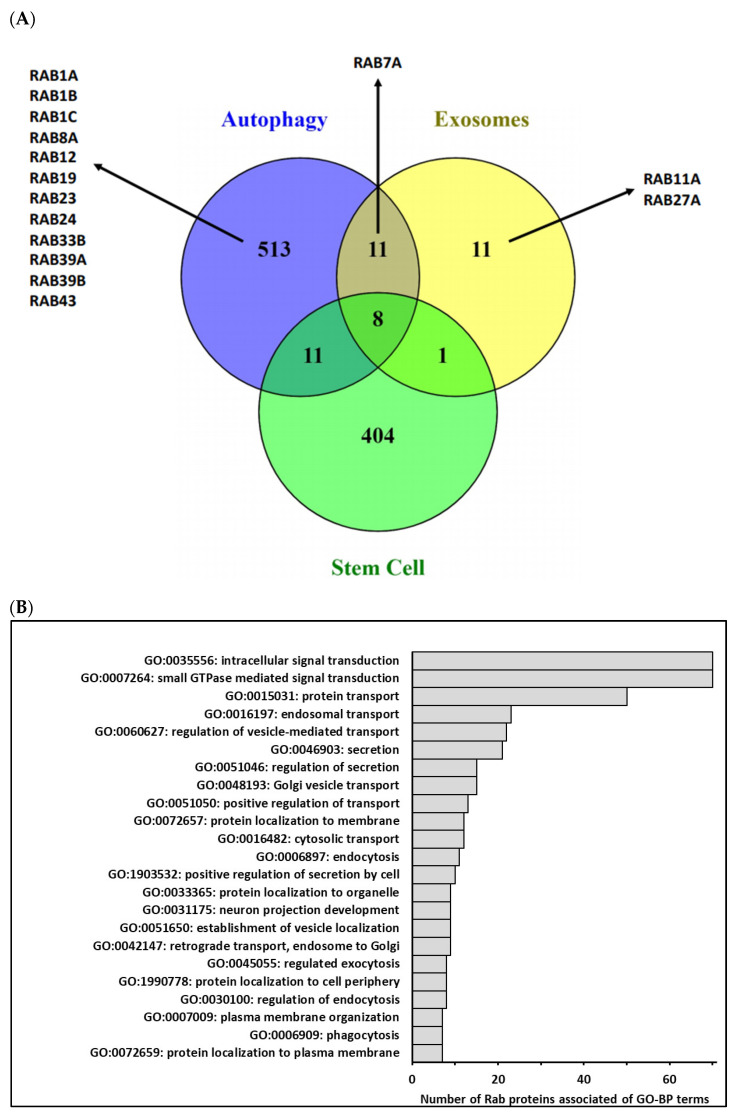
(**A**)—Comparison of human genes associated with Gene Ontology terms related to autophagy, extracellular vesicles and/or exosomes biogenesis and stem cells biological process (http://geneontology.org/ and https://www.ebi.ac.uk/QuickGO/, accessed on 26 March 2021) using Venn diagram. In fact, 543 different genes were categorized into GO-BP related to autophagy (GO:0006914 and GO:0010506), 31 different genes were categorized into GO-BP related to extracellular vesicles/exosomes (GO:0140112, GO:0097734, GO:0099156, GO:1990182, GO:0071971 and GO:1903551) and 424 different genes were categorized into GO-BP related to stem cells (GO:0019827, GO:0017145, GO:0048863, GO:0072089 and GO:0048864). Few genes were commonly associated with three biological process, and among Rab identified in these processes, only Rab7A was common to 2 processes. (**B**)—Main terms of Gene Ontology-Biological Process (GO-BP) associated with Homo Sapiens Rab proteins. GO-BP terms associated with the 70 human Rab proteins were retrieved from the DAVID database (https://david.ncifcrf.gov/ accessed on 26 March 2021) by focusing on level 5 of the GO hierarchy. Among 81 enriched terms (*p* adjusted ≤ 0.05), 33 terms were associated with more than 10% of Rab. After removing redundant GO terms using REVIGO (http://revigo.irb.hr/ accessed on 26 March 2021) according to functional similarity, we selected a short list of 23 terms summarized in the above graphic, illustrating the functional diversity of the Rab protein family.

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
