# Peer review of "Autophagy and Extracellular Vesicles, Connected to rabGTPase Family, Support Aggressiveness in Cancer Stem Cells"

_cells, 2021, doi:10.3390/cells10061330_

Round 1

Reviewer 1 Report

This review addresses an interesting and relevant topic in cancer trying to link Rab GTPases family, extracellular vesicles, autophagy and cancer stemness. The review is novel and timely however is lengthy from time to time and not clear many times. In addition, this review is very ambitious because aims to summarize 4 big topics in one review englobing all cancer types.

Beside this, my main concern is the cancer stem cell part. The CSCs are difficult to identify and very few publications really have defined well these cells. Many studies that work with for example with cell lines are based on previous publications data (e.g. markers pre-established for mouse samples), sphere formation or other experiments that do not really define if these cancer cells are real cancer stem cells or progenitors or cancer-stem like aggresive cells. This review cites many publications (e.g. 39) that have not proved enough these characteristics. So the authors really need to address well this issue before publication.

Minor points:

  • The two sides of autophagy in cancer are explained twice.
  • In the introduction, the authors need to cite relevant papers that have identified cancer stem cells as Malanchi et al, Nature 2009.
  • Sentences 113-115 are not clear.
  • Conclusion is difficult to follow, not a clear idea behind.
  • Figure 1, not clear what the blue colour means.
  • LC3 first time mentioning needs to write Microtubule-associated protein 1A/1B-light chain 3
  • I would recommend the authors to avoid using (…). It is used in many cases and it is confusing.
  • Quality of figure 2b is very low.
  • Some sentences are too long (e.g. the first part of the conclusion).

Author Response

In order to be easily saw and spotted, all changed text was written in red in the manuscript.

Reviewer 1 :

This review addresses an interesting and relevant topic in cancer trying to link Rab GTPases family, extracellular vesicles, autophagy and cancer stemness. The review is novel and timely however is lengthy from time to time and not clear many times. In addition, this review is very ambitious because aims to summarize 4 big topics in one review englobing all cancer types.

Beside this, my main concern is the cancer stem cell part. The CSCs are difficult to identify and very few publications really have defined well these cells. Many studies that work with for example with cell lines are based on previous publications data (e.g. markers pre-established for mouse samples), sphere formation or other experiments that do not really define if these cancer cells are real cancer stem cells or progenitors or cancer-stem like aggresive cells. This review cites many publications (e.g. 39) that have not proved enough these characteristics. So the authors really need to address well this issue before publication.

Response :

We perfectly agree with the reviewer concerning the lack of well define criteria to identify real cancer stem cells and specially to differentiate from progenitors. To our mind, actually, two categories of CSC characteristics could be used: i) markers expression and ii) functional properties. In the introduction, we added factual elements allowing characterization of cancer stem cells:

CSCs share with physiological stem cells self-renewal, unlimited proliferation rate and multipotent capacities. The first characterization of CSC relied on their ability to regrow a tumor in vivo (1).  However, even today, CSCs remain difficult to characterize and to isolate. Based on the same rationales as for other cellular sub-populations, CSCs are thought to be identified by the expression of specific markers. Indeed, membranal proteins such as CD133, CD44, LGR5 or intracellular actors -mainly transcriptional factors- i.e. BMI1, Oct4, Nanog, sox family were described to be enhanced in stemness (5,6). Nevertheless, unlike for others cellular subset, it is not possible to establish a sound link between CSCs and unequivocal stemness marker. That explains why, scientifics also consider functional properties to define this peculiar population. Therefore, clonogenic faculty, chemotherapeutic resistance, metastasic propension (7) as well as quiescent stage and drug efflux have also been evaluated to identify CSCs (8).

Minor points:

  • The two sides of autophagy in cancer are explained twice.

We thank the reviewer for this remark and we removed redundancy (a paragraph in the introduction was deleted)

  • In the introduction, the authors need to cite relevant papers that have identified cancer stem cells as Malanchi et al, Nature 2009.

Paper from these authors was added in the paragraph concerning stemness characterization (in the introduction, ref 7)

  • Sentences 113-115 are not clear.

We modified these sentences as follow, and hope we gain in clarity (lines 109-110)

Since both autophagy and EVs secretion are involved in CSCs status, we also depicted Rab GTPases implication in both mechanisms in relation to stemness.

  • Conclusion is difficult to follow, not a clear idea behind.

As requested, we rewrote the conclusion, drawing a more obvious thread. All the conclusion is now dealing with therapy targeting rab proteins, such a request being also asked by the reviewer 2. The conclusion is as follow :

This quick update on the current knowledge concerning autophagy, EVs and Rab GTPases in CSCs maintenance points new possible therapeutic strategies out, especially via targeting Rab GTPases. Indeed, even if the question of their real tumorigenicity is still debated, their intervention in key stages of the life of the cell justifies the development of therapies targeting them. Besides, this has been the topic of a quite recent review (158). This goal could be achieved by targeting of Rabs regulatory proteins (159), particularly their GEFs. A team designed nine peptides that could target Rab GTPases through their different interactions (160), which was done by Mitra and coworkers in 2017 using stapled peptides inhibitors of Rab25 (161) in various cell lines. This might also be achieved by the means of chemical agents such as Nexinhibs (Neutrophil-exocytosis inhibitors), which have been found to be inhibitors of the interaction between Rab27a and its effector JFC1 i.e. synaptotagmin-like protein1 (162). Thus, even if these innovative therapies targeting Rab proteins are promising, further studies would have to be undertaken. Indeed, depending on cell type, tumoral stade/grade and even the cancer type, the targeted Rab might not be the same, leading to personalized patient care.

Figure 1, not clear what the blue colour means.

We thank the reviewer for this comment. Indeed, the color and the legend were confusing. So we changed organelles color in black and consequently the legend.

  • LC3 first time mentioning needs to write Microtubule-associated protein 1A/1B-light chain 3

As requested, the LC3 abbreviation was developed when appeared for the first time (line 127)

  • I would recommend the authors to avoid using (…). It is used in many cases and it is confusing.

We modified most of the sentences where (…) were used, in order to gain clarity.

  • Quality of figure 2b is very low.

We thank the reviewer for this remark. We changed the format of the figure.

  • Some sentences are too long (e.g. the first part of the conclusion).

As previously mentioned, we rewrote the conclusion, hopping be clearer

Reviewer 2 Report

According to the title this review is regarding the role of Rab GTPases in autophagy and EVs and how this affects CSCs. The introduction to the reasoning for performing the review and the schematic image are good. However, beyond that the storyline is a little disconnected with regards to the specific roles of Rab GTPases in the processes and how targeting these Rabs would affect CSC function. Firstly there needs to be a description of the processes and the proteins involved as shown in the image (the reader may not know the pathways involved in autophagy/ EVs, therefore just specifying certain proteins in the text has no meaning unless the reader understands the roles of these proteins), then describe how Rab GTPases affect this process, then what is the evidence that this affects CSC function. The authors have too much emphasis on the roles of autophagy and EVs on CSC function but this is not the title, there is no connection to the Rabs. Also there is no description of CSC biology therefore line 136 – 138 has no relevance to a reader who does not know these markers. Sometimes treatments are described, without a short description of their action i.e line 72/73 hydroxy chloroquine – this sentence has no relevance to the text if the reader does not know what the drug is, this needs checking throughout.

The english grammar throughout is a little odd and needs to be checked thoroughly, sometimes it appears non-scientific. For example changes in the abstract need to be;

Line 14; Autophagy and EV secretion are cellular processes that help CSC survival

Line 15: Autophagy is a recycling …..

Line 17: The roles of both have been well described in the maintenance of stemness

Line 17: A common pathway involved in these processes is vesicular trafficking.

Line 33: The stochastic model suggests ….

Line 36: However, recently, a concept regarding CSCs has been evolving, that no longer considers CSCs as a static entity, but rather as a continuum, constantly sprouting and adapting to changes in the micro-environment.

Line 39: Thus, instead of having a unique CSC clone, tumors are composed of several  CSC microstates, reflecting the high heterogeneity of the tumor.

The title shouldn't have capitals - these are only necessary when the abbreviation is first mentioned in the text.

Author Response

In order to be easily saw and spotted, all changed text was written in red in the manuscript.

According to the title this review is regarding the role of Rab GTPases in autophagy and EVs and how this affects CSCs. The introduction to the reasoning for performing the review and the schematic image are good. However, beyond that the storyline is a little disconnected with regards to the specific roles of Rab GTPases in the processes and how targeting these Rabs would affect CSC function.

As asked by the reviewer, we added precisions concerning the role of Rab GTPases in: 1) Autophagy (lines 106-113 ; 116-118) 2) EVs secretion (lines 242-248) 3) CSCs behavior (lines 348-350), especially EMT  (lines 424-434). Elements concerning the special topic of Rab targeting is also exposed in conclusion of the manuscript (lines 463-475).

Firstly there needs to be a description of the processes and the proteins involved as shown in the image (the reader may not know the pathways involved in autophagy/ EVs, therefore just specifying certain proteins in the text has no meaning unless the reader understands the roles of these proteins), then describe how Rab GTPases affect this process, then what is the evidence that this affects CSC function.

In order to gain clarity, we shortened the introduction and, at the beginning of each section (Autophagy -lines 92-103- and EVs-lines 232-237), we added a brief description of the process and gave references where to found more details.

The authors have too much emphasis on the roles of autophagy and EVs on CSC function but this is not the title, there is no connection to the Rabs. Also there is no description of CSC biology therefore line 136 – 138 has no relevance to a reader who does not know these markers.

We paid attention to this remark and we added in the introduction a description of both CSCs markers and functional characteristics (lines 30-45)

Sometimes treatments are described, without a short description of their action i.e line 72/73 hydroxy chloroquine – this sentence has no relevance to the text if the reader does not know what the drug is, this needs checking throughout.

Precisions concerning chloroquine was brought out (line 150) and the same for some other pharmacological molecules (lines 161 ; 313).

The english grammar throughout is a little odd and needs to be checked thoroughly, sometimes it appears non-scientific. For example changes in the abstract need to be;

Line 14; Autophagy and EV secretion are cellular processes that help CSC survival

Line 15: Autophagy is a recycling …..

Line 17: The roles of both have been well described in the maintenance of stemness

Line 17: A common pathway involved in these processes is vesicular trafficking.

As requested, we changed the 4 sentences of the abstract

Line 33: The stochastic model suggests ….

Du to response to another reviewer, this sentence is deleted

Line 36: However, recently, a concept regarding CSCs has been evolving, that no longer considers CSCs as a static entity, but rather as a continuum, constantly sprouting and adapting to changes in the micro-environment.

Line 39: Thus, instead of having a unique CSC clone, tumors are composed of several  CSC microstates, reflecting the high heterogeneity of the tumor.

Both sentences were modified

The title shouldn't have capitals - these are only necessary when the abbreviation is first mentioned in the text.

We removed the capitals in the titles

Reviewer 3 Report

Brunel et al. have written a comprehensive review on the role of the autophagic pathway and exosome secretion on the formation and maintenance of cancer stem cells (CSC). They connect these two cellular processes through their regulation by Rab GTPases. Finally, they propose that targeting Rab GTPases may be a successful strategy to inhibit CSC survival and thus therapeutic resistance. The review is interesting and timely, however I think it is necessary to address a number of weaknesses prior to publication.

Major Points

  • Given the focus of this review is the role of Rab GTPases in determining the aggressiveness of cancer stem cells, there is actually not very much information about Rab GTPases. I would like to see the authors provide more detail about the roles Rab GTPases play in regulating autophagy and exosome secretion.
  • The authors should discuss potential approaches to targeting Rab GTPases (e.g. small molecule vs genetic targeting), and the challenges that are likely to be encountered. Mention if Rabs have been successfully targeted in other diseases.
  • The authors should discuss whether any Rab GTPases been found to be upregulated in cancer stem cells? Alternatively, have any Rabs been implicated in regulating epithelial-mesenchymal transition?
  • The legend for Figure 2 requires more information.

Minor Points

  • Line 14: Change to ‘They are able to survive harsh conditions….’
  • Line 15: Change to ‘Among the processes promoting CSC survival are autophagy and extracellular vesicle (EV) secretion. Autophagy is a recycling process and EV secretion is essential…..’
  • Line 18: ‘the involvement of vesicular trafficking, and thus regulation by Rab GTPases. In this Review we analyse the role played by Rab GTPases in stemness status, either directly or through their regulation of autophagy and EV secretion.’
  • Line 40: ‘reflecting the high tumor heterogeneity.’
  • The authors should expand the definition of autophagy to something like ‘a mechanism for the degradation and recycling of cellular components’. Calling it just a recycling pathway risks readers confusing it with the endosomal recycling pathway.
  • Line 54: Change to ‘long-lived’
  • Line 89: Change to ‘Indeed, in many cancer types, a difference in EV content between healthy persons and patients with cancer has been observed’
  • Line 102: should be ‘guanine-nucleotide exchange factors (GEFs)’
  • Line 110: Change to ‘In this review, we focused on the role played by Rab GTPases in determining CSC status.’
  • Lines 120 and 238: ‘involves’ instead of ‘implies’
  • Line 282: ‘promoted the formation of..’ instead of ‘gave birth to..’

Author Response

In order to be easily saw and spotted, all changed text was written in red in the manuscript.

Reviewer 3 :

Brunel et al. have written a comprehensive review on the role of the autophagic pathway and exosome secretion on the formation and maintenance of cancer stem cells (CSC). They connect these two cellular processes through their regulation by Rab GTPases. Finally, they propose that targeting Rab GTPases may be a successful strategy to inhibit CSC survival and thus therapeutic resistance. The review is interesting and timely, however I think it is necessary to address a number of weaknesses prior to publication.

Major Points

  • Given the focus of this review is the role of Rab GTPases in determining the aggressiveness of cancer stem cells, there is actually not very much information about Rab GTPases. I would like to see the authors provide more detail about the roles Rab GTPases play in regulating autophagy and exosome secretion.

We thank the reviewer for this pertinent remark. We added more informative content concerning the involvement of Rab GTPases in autophagy regulation :

“One of the first Rab-GTPase described in autophagy regulation was Rab33B in 2008, a Golgi-associated Rab protein (38) via its interaction with the ATG16L protein, one of the components of a proteic complex involved in autophagosome elongation. Additionally, Rab7 might be one of the most described Rab-GTPase protein in the autophagic pathway regulation. Indeed, this particular protein plays an important role in lysosome biosynthesis and function (39). This is why autophagosome fusion with the lysosome has been, in part, attributed to Rab7, particularly to Rab7a (40). Moreover, Rab1 seems to participate to the phagophore formation by regulating Atg9 localization (41,42), whereas Rab32 is required for the autophagosome formation (43,44) and Rab5 for its closure (25). Besides, Rab24 has been shown to be co-localized with Microtubule-associated protein 1A/1B-light chain 3 (MAP-LC3) (45). Furthermore, Rab2 has recently been shown as involved in the regulation of autophagosome and autolylosome formation in different mammalian cell lines (46).»

And in EVs secretion

  • There are mainly three Rab-GTPases described in EVs biogenesis and secretion (96): Rab11, Rab35 and Rab27A. All those three proteins are implicated in the docking and fusion of the MultiVesicular Bodies (MVBs). Rab11 activity has been shown to be Calcium-dependent in the erythroleukemia cell line K562 in 2005 (97). More recently, the promotion of exosome secretion by Rab35 has been observed in hepatocellular carcinoma (98), when it had been previously demonstrated few years earlier in Oli-neu cell line i.e. oligodendroglial precursor (99).
  • The authors should discuss potential approaches to targeting Rab GTPases (e.g. small molecule vs genetic targeting), and the challenges that are likely to be encountered. Mention if Rabs have been successfully targeted in other diseases.

We thank the reviewer for this remark. We treated this particular interesting point in the conclusion.

This quick update on the current knowledge concerning autophagy, EVs and Rab GTPases in CSCs maintenance points new possible therapeutic strategies out, especially via targeting Rab GTPases. Indeed, even if the question of their real tumorigenicity is still debated, their intervention in key stages of the life of the cell justifies the development of therapies targeting them. Besides, this has been the topic of a quite recent review (158). This goal could be achieved by targeting of Rabs regulatory proteins (159), particularly their GEFs. A team designed nine peptides that could target Rab GTPases through their different interactions (160), which was done by Mitra and coworkers in 2017 using stapled peptides inhibitors of Rab25 (161) in various cell lines. This might also be achieved by the means of chemical agents such as Nexinhibs (Neutrophil-exocytosis inhibitors), which have been found to be inhibitors of the interaction between Rab27a and its effector JFC1 i.e. synaptotagmin-like protein1 (162). Thus, even if these innovative therapies targeting Rab proteins are promising, further studies would have to be undertaken. Indeed, depending on cell type, tumoral stade/grade and even the cancer type, the targeted Rab might not be the same, leading to personalized patient care.

  • The authors should discuss whether any Rab GTPases been found to be upregulated in cancer stem cells? Alternatively, have any Rabs been implicated in regulating epithelial-mesenchymal transition?

Those are very interesting questions. After further research, complementary information have been provided in the introduction concerning Rabs implicated in EMT.

“In addition, various Rabs have been attributed a role in cancer cells migration and Epithelial-to-Mesenchymal Transition (EMT). Among them, Rab23 was found to be responsible for EMT in ovarian cancer and hepatocellular carcinoma (30,31). Also, Rab11 has been described as a regulator of E-cadherin, thus promoting cell migration in colorectal cancer (32,33). On the contrary, some Rab-GTPases negatively regulate the EMT and their silencing has been proven to induce it. This is notably the case concerning the epigenetic silencing of Rab39A in cervical cancer (34) or the downregulation of Rab17 in non-small cell lung carcinoma (35).”

Concerning the expression levels of Rabs in CSCs, it has been added at the beginning of the “Rab GTPases and CSCs” chapter

“Moreover, on overexpression of Rab5b in breast cancer stem cells has been highlighted indirectly in a 2018 study (141). Indeed, authors showed that miR-130a-3p, involved in Rab5b regulation, is downregulated in such CSCs.”

  • The legend for Figure 2 requires more information.

We thank the reviewer for this remark. We added further explanations to the legend :

Figure 2. A – Comparison of human genes associated to Gene Ontology terms related to autophagy, extracellular vesicles and/or exosomes biogenesis and stem cells biological process (http://geneontology.org/ and https://www.ebi.ac.uk/QuickGO/ ) using Venn diagram. Actually, 543 different genes were categorized into GO-BP related to autophagy (GO:0006914 and GO:0010506), 31 different genes were categorized into GO-BP related to extracellular vesicles/exosomes (GO:0140112, GO:0097734, GO:0099156, GO:1990182, GO:0071971 and GO:1903551) and 424 different genes were categorized into GO-BP related to stem cells (GO:0019827, GO:0017145, GO:0048863, GO:0072089 and GO:0048864). Few genes were commonly associated to three biological process, and among Rab identified in these processes, only Rab7A was common to 2 process. B – Main terms of Gene Ontology-Biological Process (GO-BP) associated to Homo Sapiens Rab proteins. GO-BP terms associated to the 70 human Rab proteins were retrieved from DAVID database (https://david.ncifcrf.gov/) by focusing on level-5 of GO hierarchy. Among 81 enriched terms (p adjusted ≤ 0.05), 33 terms were associated to more of 10% of Rab. After removing redundant GO terms using REVIGO (http://revigo.irb.hr/) according functional similarity, we selected a short list of 23 terms resumed in the above graphic, illustrating functional diversity of Rab protein family.

Minor Points

We thank the reviewer for these comments concerning the remaining mistakes with the English language. All the changes have been made, and corresponding lines of the manuscript are indicated :

  • Line 14: Change to ‘They are able to survive harsh conditions….’ Line 14
  • Line 15: Change to ‘Among the processes promoting CSC survival are autophagy and extracellular vesicle (EV) secretion. Autophagy is a recycling process and EV secretion is essential…..’ lines 14,15,16
  • Line 18: ‘the involvement of vesicular trafficking, and thus regulation by Rab GTPases. In this Review we analyse the role played by Rab GTPases in stemness status, either directly or through their regulation of autophagy and EV secretion.’ Lines 18,19
  • Line 40: ‘reflecting the high tumor heterogeneity.’ Line 45
  • The authors should expand the definition of autophagy to something like ‘a mechanism for the degradation and recycling of cellular components’. Calling it just a recycling pathway risks readers confusing it with the endosomal recycling pathway.

As requested, authophagy has been defined in a clearer way in order to not confuse it with the endolysosomal pathway.

“It is a highly conserved degradation and recycling mechanism of cellular components, complementary to the proteasome” lines 56,57

  • Line 54: Change to ‘long-lived’ line 58
  • Line 89: Change to ‘Indeed, in many cancer types, a difference in EV content between healthy persons and patients with cancer has been observed’ lines 79,80
  • Line 102: should be ‘guanine-nucleotide exchange factors (GEFs)’ line 91
  • Line 110: Change to ‘In this review, we focused on the role played by Rab GTPases in determining CSC status.’ Line 106
  • Lines 120 and 238: ‘involves’ instead of ‘implies’ lines 114 and 241
  • Line 282: ‘promoted the formation of..’ instead of ‘gave birth to..’ line 291

Round 2

Reviewer 1 Report

In my opinion the authors are not still accurate when they call cancer stem cells to cell types that have been described elsewhere as cancer stem cells but are not.

Author Response

We thank the reviewer for this comment. Even if we agree with him/her for the most time on the vagueness of the definition of stemness, we have reviewed the articles as they were published and where they referred to cancer stem cells, we have stated these cell states as the authors of the articles themselves described them, without questioning their definitions. 

In order to warn readers, we add the sentence of the reviewer:

“However, that do not really define if these cancer cells are real cancer stem cells or progenitors or cancer-stem like aggressive cells.”

Reviewer 2 Report

Although the concerns have been addressed re adding more details about Rab GTPases in EV and autophagy, the link between the roles Rabs play in these processes and how they specifically contribute CSC aggressiveness has not been expanded. It maybe that there is no more information than the authors have stated here, but as the roles of autophagy and EV in CSC aggressiveness are not the focus of the title there is far too much detail in these sections (which just needed to be summarised in order to bring in Rab GTPases as the main focus). 

The grammar still needs checking throughout, even in the additional text. I had previously only given examples.

Author Response

This comment deeply wondered us and led us to propose a slightly different title. If you think it will be more acceptable, the tittle could be change in: “Autophagy and extracellular vesicles, connected to rabGTPase family, support aggressiveness in cancer stem cells.”

Nevertheless and as mentioned by the reviewer, the topic of this review is not extensively studied, that's what makes it interesting and what prompted us to make it the state of the art. When we cross “rab gtpase and CSC” in pubmed, there are only 5 publications.

In order to take into account the reviewer’s comments, we shorten the text; the removed sentences appear striped in the new text and modifications are underlined.

The text was checked by an English native mother tongue and we hope it will improve the quality of the language.

Reviewer 3 Report

I would like to thank the authors for addressing my comments and the manuscript has been improved by the changes that they made.

Author Response

Thank you for your comments

Round 3

Reviewer 1 Report

The authors have not addressed my concern about cancer stem cells. They have added a sentence when citing ref 39 but there are other many references with the same problem.

As the authors state that they believe what other authors write defining "cancer stem cells", then they should write this at the beginning of the review (for example in the introduction). This is the way to tell the readers that when the authors write "cancer stem cells" they have not verified if these are real cancer stem cells.

In my opinion, a review is not just to do a summary and repeat the words of what others have done but this is up to authors and editors opinions.

Author Response

As requested by the reviewer we added a warning to readers in the introduction:

“It is important to be aware that in many studies, including those cited in this review, the tumorigenicity requirement, which should be unavoidable, is not always verified. Readers, by referring to the cited publications, will be able to make up their own minds of stemness.”